# Co-creating safe spaces: Study protocol for translational research on innovative alternatives to the emergency department for people experiencing emotional distress and/or suicidal crisis

**Michelle Banfield**[1], **Scott J. Fitzpatrick**[1]*, **Heather Lamb**[1], **Melanie Giugni**[1], **Alison L. Calear**[1], **Erin Stewart**[2], **Maree Pavloudis**[2], **Lucy Ellen**[3], **Ginny Sargent**[4], **Helen Skeat**[4], **Bronwen Edwards**[5], **Benn Miller**[6], **Amelia Gulliver**[1], **Louise A. Ellis**[7], **Vida Bliokas**[8], **Purity Goj**[9], **Melissa Lee**[9], **Kelly Stewart**[10], **Glenda Webb**[6], **Merkitta Main**[11], **Carrie Lumby**[12], **Kelly Wells**[13], **Carolyn McKay**[14], **Philip J. Batterham**[1], **Alyssa R. Morse**[1‡], **Fiona Shand**[14‡], **Stride Safe Space Team**[15¶]

1 Centre for Mental Health Research, The Australian National University, Australian Capital Territory, Australia, 2 ACT Mental Health Consumer Network, Australian Capital Territory, Australia, 3 Centre for Social Research & Methods, The Australian National University, Australian Capital Territory, Australia, 4 Population Health Exchange, The Australian National University, Australian Capital Territory, Australia, 5 Roses in the Ocean, Brisbane, Australia, 6 Towards Zero Suicides Initiatives, South Western Sydney Local Health District, Sydney, Australia, 7 Australian Institute of Health Innovation, Macquarie University, Sydney, Australia, 8 School of Psychology, University of Wollongong, Wollongong, Australia, 9 ACT Health Directorate, Australian Capital Territory, Australia, 10 Sonder, Adelaide, Australia, 11 South Western Sydney Local Health District, Sydney, Australia, 12 Illawarra Shoalhaven Suicide Prevention Collaborative, Wollongong, Australia, 13 Adelaide Primary Health Network, Adelaide, Australia, 14 Black Dog Institute, University of New South Wales, Sydney, Australia, 15 Stride, Sydney, Australia

‡ ARM and FS authors are joint last authors on this work.
¶ Membership of the Stride Safe Space Team is provided in the Acknowledgments.
* scott.fitzpatrick@anu.edu.au

**Data Availability Statement:** No datasets were generated or analysed during the current study.

## Abstract

### Introduction

Safe spaces are an alternative to emergency departments, which are often unable to provide optimum care for people experiencing emotional distress and/or suicidal crisis. At present, there are several different safe space models being trialled in Australia. However, research examining the effectiveness of safe space models, especially in community settings, is rare. In this paper, we present a protocol for a study in which we will investigate the implementation, effectiveness, and sustainability of safe space models as genuine alternatives for people who might usually present to the emergency department or choose not to access help due to past negative experiences.

### Material and methods

We will use a mixed methods, co-designed study design, conducted according to the principles of community-based participatory research to obtain deep insights into the benefits of

Ethical restrictions will apply to the sharing of the de-identified dataset generated in this study as it will contain potentially identifying or sensitive information. However, de-identified and aggregate data will be made available to participating sites to facilitate future evaluation and quality improvement activities. Data will be archived in the Australian Data Archive, with access restricted to representatives from participating sites by request to the Principal Investigator or her nominated delegate.

**Funding:** This project is funded by an Australian Government Department of Health National Suicide Prevention Research Fund Targeted Research Grant, managed by Suicide Prevention Australia. ARM is funded by an Australian Government Department of Health National Suicide Prevention Research Fund Post-Doctoral Fellowship, managed by Suicide Prevention Australia. PJB is supported by NHMRC Fellowship 1158707 and AC is supported by NHMRC Fellowship 1173146. The funders have no role in study design, data collection and analysis, decision to publish or preparation of the manuscript.

**Competing interests:** The authors have declared that no competing interests exist.

different safe space models, potential challenges, and facilitators of effective practice. We developed the study plan and evaluation framework using the RE-AIM framework, and this will be used to assess key outcomes related to *reach*, *effectiveness*, *adoption*, *implementation*, and *maintenance*. Data collection will comprise quantitative measures on access, use, satisfaction, (cost) effectiveness, distress, and suicidal ideation; and qualitative assessments of service implementation, experience, feasibility, acceptability, community awareness, and the fidelity of the models to service co-design. Data will be collected and analysed concurrently throughout the trial period of the initiatives.

## Discussion

This study will enable an extensive investigation of safe spaces that will inform local delivery and provide a broader understanding of the key features of safe spaces as acceptable and effective alternatives to hospital-based care for people experiencing emotional distress and/or suicidal crisis. This study will also contribute to a growing body of research on the role and benefits of peer support and provide critical new knowledge on the successes and challenges of service co-design to inform future practice.

## Introduction

In Australia, there is increasing government investment in 'safe spaces' for people experiencing emotional distress and suicidal crisis. First implemented in the UK [1], safe spaces are an alternative to emergency departments (EDs), which are often poorly equipped to support persons in emotional distress and to provide high quality care commensurate to need [2–4]. Service users report experiencing negative attitudes and behaviour from emergency department staff, including, humiliation, discrimination, a lack of empathy, and denial of routine care [4, 5]. In many instances, contact with EDs has been shown to perpetuate a cycle of shame, distress, and further self-harm, while discouraging future help-seeking [5–7].

Inadequate capacity in Australia's mental health system has resulted in a growing number of presentations to ED for crisis care and support, often because mental health services are unable to provide alternatives to EDs for people in extreme distress [8, 9]. For ED staff providing care to people experiencing suicidal crisis, numerous barriers exist to the provision of optimum care. Environmental and systemic aspects of the ED such as limited resources, privacy, the multidisciplinary nature of care, and procedures for suicide risk assessment, coupled with increasing levels of stress and staff burnout, have led to questions about the suitability of EDs for those experiencing emotional distress and/or suicidal crisis [10, 11].

Non-clinical spaces where people in crisis can receive appropriate support from peers aim to overcome these problems [12]. Non-clinical approaches are those that do not involve any type of medical diagnosis or treatment processes. The emergence of non-clinical approaches to suicide prevention coincides with the international expansion of peer support within community mental health [13]. Underpinning characteristics of peer support models in suicide prevention include the application of lived experience and strength-focused social and practical support to foster hope, connection, and healing to those in crisis [14].

To date, very little research has been conducted in Australia or globally to assess the effectiveness of non-clinical alternatives to EDs and hospital-based care for people experiencing suicidal crisis or distress. Emerging research on outpatient, peer-led services suggests that

non-clinical care settings are perceived as helpful and positive by those who use them, as well as by those working in health and community services [15]. Peer workers can facilitate better experiences by practising positive role modelling and building strong and trusting relationships with people experiencing distress [16, 17]. The use of non-clinical environments and peer support are effective in helping to alleviate crises and constitute a cost effective alternative to EDs [12, 18].

The proposed research will determine the feasibility and effectiveness of safe space models as genuine alternatives for people experiencing emotional distress and/or suicidal crisis who might usually present to the ED or choose not to access help due to past negative experiences. At present, there are several different safe space models being developed and trialled in Australia. In the current study we will compare six approaches across three Australian states and territories, which will provide valuable insight into the benefits of each of these models, potential challenges, and facilitators of effective practice.

## Research design

"Co-creating safe spaces" is a mixed methods study that will be conducted according to the principles of community-based participatory research [19]. While the importance of such an approach is acknowledged as best practice in Australian national policy documents [20–22], true co-design and partnership is rarely implemented. Co-design is core to both the safe spaces themselves and the research study, building a culture of equal and shared knowledge across all stakeholder groups involved. This includes a particular focus on the experiences of those who access the safe spaces for support (termed safe space guests) and peer support workers. Members of the core project team, which includes researchers with and without their own lived experience of mental health issues and/or suicide; site partners including decision-makers, managers, and peer workers; and people from the community with lived experience, will co-design the final evaluation plan, support further lived experience and peer worker involvement, participate in data collection and interpretation of findings, and contribute to the dissemination and implementation of results of the research.

The study design uses an implementation science approach. The RE-AIM framework [23] will be used for measuring key outcomes related to *reach*, *effectiveness*, *adoption*, *implementation*, and *maintenance*.

### Aims

The specific aims of the study are as follows:

1. Evaluate the *reach*, accessibility, and ability of safe spaces to meet the needs of guests as an alternative to the ED.

2. Assess the *effectiveness* of safe spaces and the impact of co-design processes for people experiencing a crisis and for those assisting them.

3. Explore *adoption* of the co-designed models in local health and emergency service systems.

4. Evaluate the successes and challenges of *implementation* of the safe spaces in each jurisdiction with a focus on the impacts of co-design.

5. Assess *maintenance*, including sustainability and cost-effectiveness of the locally implemented models. Sustainability incorporates the data and processes needed to continually improve the experiences and outcomes of guests, and the processes needed to engage those in the community who may not seek help during a crisis.

## Setting and participants

This is a multi-site study of safe spaces being conducted by the Centre for Mental Health Research, The Australian National University in collaboration with Roses in the Ocean (a leading Australian lived experience of suicide organisation), the steering committees implementing the safe spaces in each jurisdiction, The Black Dog Institute (University of New South Wales), The Australian Institute of Health Innovation (Macquarie University), and the University of Wollongong. The study will be conducted across six trial sites in three states and territories: Australian Capital Territory (ACT), New South Wales (NSW), and South Australia (SA).

There are five key stakeholder groups for the study who will form the core participant groups for the research:

1. People accessing the safe spaces for support (safe space guests)

2. Safe space staff

3. Hospital and health services staff

4. Community of potential safe space guests

5. Safe space steering groups, partners, and co-design participants.

## Lived experience involvement

People with lived experience of suicidal crisis and/or attempt are integrated into the project and engaged in multiple ways to ensure respectful and effective involvement. The lead investigator and author has lived experience of suicidal ideation, including accessing an ED for suicidal crisis, and several co-investigators have experience caring for someone with suicidal thoughts and behaviour. The team includes three additional lived experience investigators, two associated with universities and one with Roses in the Ocean. The initial evaluation proposal was drafted with the ACT Safe Haven Café Steering Committee and refined with input from other partner organisations (Roses in the Ocean, the Illawarra Shoalhaven Suicide Prevention Collaborative, the ACT Mental Health Consumer Network, and Stride). All investigators are responsible for supporting project co-production processes and will collaborate with site partners and people with lived experience of suicidal crisis across all stages of the research. The team will monitor ways of working to ensure that we are true to the principles agreed to at the outset, using process evaluation tools (for example, the Collaboration Health Assessment Tool) [24]. This embeds lived experience leadership and feedback loops within all research stages, from conception through to dissemination. The participatory research design promotes stakeholder engagement with the research processes and provides strong connections with each key stakeholder group.

The study co-design was developed to be flexible, and to provide individual sites with a choice of study components and data collection tools for evaluating key RE-AIM domains of interest for local purposes, while also contributing to the overarching translational research project. The chosen study components, outcome measures, and methods have all been developed and/or validated by stakeholders (service providers and lived experience representatives) to assess burden and acceptability of processes before implementation of the research, and to minimise the possibility that key areas of importance for the research are missed.

## Study components

The research design comprises multiple study components (see **Table 1**). These are as follows:

**Table 1. Study design.**

| Study component | Data Collection Method | RE-AIM Domains |
|---|---|---|
| **Guest study** | Guest questionnaires: Entry/Exit Survey; Evaluation Wheel; Graffiti Wall; Online Survey | Effectiveness |
| | Guest Journeys: Health Administrative Data; Unstructured Interviews; Body Mapping | Effectiveness |
| | | Maintenance |
| **Staff study** | Safe Space Staff: Staff Online Survey; Focus Groups or Semi-Structured Interviews | Effectiveness |
| | | Adoption |
| | | Implementation |
| | | Maintenance |
| | Hospital and Health Services Staff: Staff Online Survey; Focus Groups | Effectiveness |
| | | Adoption |
| | | Implementation |
| | | Maintenance |
| **System and economic study** | Health Administrative Data: Emergency Department Data; Costings | Reach |
| | | Effectiveness |
| | | Implementation |
| | | Maintenance |
| **Community study** | Community Online Survey | Reach |
| | | Maintenance |
| **Service co-design study** | Document Analysis; Co-design Online Survey; Focus Groups or Semi-Structured Interviews | Implementation |

- Guest study

- Staff study

- System and economic study

- Community study

- Service co-design study

## Materials and methods

### Ethics

It is not anticipated that people with known impaired capacity will participate in the project. The study recognises young people aged 16 years and over as 'mature minors'. This aligns with the co-design model established by the safe spaces that welcomes guests aged 16 years and over, and that acknowledges mature adolescent's autonomy, decision-making capacity, and their right to access health services independently, as well as to participate in research on services designed for them. Recruitment materials and data collection tools have been co-designed by people with lived experience of suicidal crisis/attempt, their carers and support people, and health professionals, and are tailored to meet the needs of young people and those who may be experiencing distress. Participants will be encouraged to seek any further information that they require from investigators to give informed consent and can involve family members, carers, or support people if they require assistance. Participants will not be required to obtain parental/guardian consent. The Australian Capital Territory Health Human Research Ethics Committee (Reference Number 2022.ETH.00043) has approved the study protocol and consent procedures.

## Guest study

The guest study will examine whether safe spaces are a feasible and effective approach to support people experiencing varying degrees of emotional distress including a suicidal crisis. Individual sites will have the flexibility of choosing between several data collection methods for measuring guest outcomes to best meet the needs of guests and staff.

## Participants, recruitment, and consent

Safe space guests are people accessing the safe spaces for support. Guests aged 16 years and over who have presented to a safe space at one of the six trial sites will be recruited. Based on the assumption that there is an average of 20 guests per month at each of the 6 sites, the expected sample size at 19 months (after 18 months' data collection) is 2,160. With a conservative participation rate of 30%, it is expected that we will recruit 648 participants (108 from each site).

Participants will be notified of the entry/exit survey by safe space staff as part of routine guest contact, but it is important that participants understand that their consent to participate is entirely voluntary and that it will not affect the services they receive. A participant information statement will be provided to participants through a digital tablet, with paper copies available for participants to keep. Participants will be required to read the information statement and to provide informed consent before commencing the research activities at the site, including the entry/exit survey, evaluation wheel, and graffiti wall as described below.

For the online survey, participants will be notified of the survey by safe space staff and given a postcard with a QR code through which they can access the survey. Participants will be required to read a participant information statement, which will be available to download and/ or print, and to provide informed consent before commencing the survey.

Opportunity to participate in guest journeys and body mapping will be offered two ways. While at the safe space, where appropriate and interest is indicated, participants will be given a brief description of these methods by the safe space staff. Those interested will be provided the name and contact details of one of the co-investigators on the project and will be invited to contact them directly to hear more about the study. Additionally, those participating in the online survey will be asked if they consent to being contacted in the future for evaluation purposes and their preferred method of contact. An embedded link at the end of the survey will take participants to a separate webpage that collects their personal information. Those who consent will be contacted by the research team via their preferred method of contact and invited to participate in the guest journey, body mapping, or both. Potential participants will be sent a participant information statement and required to read this and complete an informed consent form prior to participating in the guest journey, interview and/or body mapping.

As part of the guest journeys, we will seek written permission from participants to access identified health administrative data on mental health-related appointments, referrals, emergency department visits, and hospitalisations. Participants will be able to participate in the interview and/or body mapping only component of the guest journey without consenting to their health administrative data being accessed.

## Data collection

*Guest questionnaires.* Guests will be invited to complete a brief entry and exit survey to assess the effectiveness of the safe space models. These surveys are intended to be unobtrusive and embedded in the routine continuous quality improvement practices of the safe spaces and will take the form of one or more of the following procedures:

i. Digital tablets placed in the entry/exit areas of the safe spaces. The entry/exit surveys will comprise three questions related to effectiveness and access.

ii. An evaluation wheel: this takes the form of a wheel divided into segments on which guests rate different outcomes related to safety, effectiveness, person-centeredness, and experience on a scale from 1–10. The evaluation wheel will be available on a digital tablet or on an A4 card.

iii. A graffiti wall: this method involves a large paper flipchart (where individual pages can be removed) on which guests can respond to questions about safety, effectiveness, person-centeredness, and experience.

To collect in-depth knowledge on the effectiveness of the safe space models, guests will be invited to participate in a longer, online survey. This survey will collect information on accessibility, use, experiences, and effectiveness of the safe space models, as well as questions about distress, suicidal ideation, and attempts.

*Health administrative data*. Data on the sociodemographic background of guests, guests' reasons for presentation, guests' distress, and guests' satisfaction levels will be collected by individual safe space sites as part of routine practice. Individual safe space sites may also collect YES (Your Experience of Service) data, a pre-existing national survey instrument designed to capture information from service users about their experiences of care [25]. Access to these data will be requested through Local Health Districts, Primary Health Networks, and Community Managed Organisations responsible for the collection of this data.

*Guest journeys*. Guest journeys will be conducted with consenting participants to provide rich qualitative information about guests and to examine service pathways including referral patterns and service use. Options include participation in an unstructured interview or body mapping activity in combination with health administrative data.

Interviews will provide an overview of the journey taken as described by the safe space guest from before entry to the service to exit, and in the subsequent six months. The data will be used to identify service issues and suggested improvements, with an emphasis on guests' needs and experiences. Drawing on grounded theory, guest journeys will involve a series of unstructured interviews exploring a guest's pathway to the service, experience of the service, and their experiences after accessing the safe space [26, 27]. Interviews will take a participant-led, informal approach similar to yarning in Aboriginal and Torres Strait Islander research. Yarning in research is an informal yet purposeful process for relating and connecting through the telling of stories [28].

As an alternative to qualitative interviews, an arts-based method will be offered to participants as a mode of participatory research to explore guests' experiences. Body mapping is a visual, narrative, and participatory methodology that combines visual and oral media. It can broadly be described as the process of creating human life-sized body images using drawing, painting, or other arts-based techniques [29]. Body mapping creates spaces for participants to convey their stories in their own terms, actively participating in the data generation process and choosing what information they consider relevant or are willing to share [29]. Body mapping will be used to engage participants in discussions of their health and wellbeing. The final outcome of the body-map storytelling process is a mapped story composed of 3 elements: i) a life size body map; ii) a key to describe each visual element found on the map; and iii) a story narrated in the first person [30].

In addition to interviews and/or body mapping, we will seek written permission from participants to access identified health administrative data on mental health-related appointments, referrals, emergency department visits, and hospitalisations. These data will be used to

create a map of the care pathway from the participant perspective. Access to health administrative data will be requested through Local Health Districts and Primary Health Networks from which participants accessed the safe space sites. Health administrative data will augment the guest journey interviews and body mapping. However, participants will be able to participate in these activities even if they do not consent to their identified health administrative data being accessed.

## Data analysis

Given the small numbers of guests accessing the safe spaces, it is expected that quantitative data will be primarily descriptive and comprise overall service scores on measures of safety, effectiveness, person-centeredness, and experience. Where possible, differences between settings and the effects of potential confounds will be tested using multi-level linear models, which will account for clustering of individuals within models and settings.

Qualitative guest journey data will be analysed using a grounded theory approach that employs simultaneous data collection and analysis to allow for unanticipated directions of inquiry that emerge through successive levels of analyses [31]. Data collection will continue until categories and relationships are saturated and new data do not add to the developing theory [32].

Analysis and interpretation of body maps will occur in two stages. The first analytical level takes place collaboratively during the creation of the body map, a time when researcher/facilitator and participant co-construct meanings focusing on the narrated story or visual keys. The second level of analysis is performed by researchers who will conduct inductive and deductive analysis for each individual body map and/or across several body maps [29].

## Staff study

The staff study will examine whether safe space models are feasible and acceptable to implement, effective at assisting guests to overcome crises, and if providing support to guests will be a positive experience for safe space staff.

**Participants and recruitment.** Safe space staff are peer support workers employed by one of the safe space sites who provide support for guests attending the safe space, service leaders (who may or may not be peers), and associated clinical and service management staff. Hospital and health services staff are persons who work in the emergency department or in mental health inpatient and outpatient settings within the local health districts/networks where safe spaces are located. Safe space, hospital, and health services staff aged 18 years and over will be recruited.

Safe space, hospital, and health services staff will be invited to participate by members of the research team during project co-design and throughout study implementation via the site partner contacts, who will be asked to circulate an invitation to participate in the online survey via an email sent through the staff intranet and relevant staff mailing lists. Participants will be required to read a participant information statement and to provide informed consent before commencing the survey. Staff who participate in the online survey will be asked if they consent to being contacted in the future for evaluation purposes and their preferred method of contact. An embedded link at the end of the survey will take participants to a separate webpage that collects their personal information. Those who consent will be contacted by the research team via their preferred method of contact and invited to participate in a focus group or interview. A recruitment email specifically for the focus groups/interviews will also be circulated via the staff intranet and relevant staff mailing lists, as staff members who do not participate in a survey may still like to participate in a focus group/interview. Interested staff members will be

asked to contact the research team via the details provided in the recruitment email. Participants will be sent a participant information statement and required to read this and complete an informed consent form prior to participating in the focus group/interview. A reminder email will be sent at 2-weeks if there is no response to the first email.

**Data collection.** Data collection tools for the staff study include interview and focus group protocols and survey questions. These will be developed from the Consolidated Framework for Implementation Research [33].

*Safe space staff.* Online Survey: An online survey will be conducted with safe space staff to explore the feasibility and acceptability of safe space models regarding barriers and facilitators to their adoption, and any impacts the safe space sites have had on workplace culture and attitudes towards suicide.

Focus Groups or Semi-Structured Interviews: Focus Groups will be conducted with safe space staff. The aim of these is twofold. First, to examine the experiences of safe space staff in providing support to people in distress or crisis; and second, to gather information on the different elements of the safe space models for managing the guest journey and how these can be used to maximise/improve the focus of care to best meet guests' needs. This information will be used to process map the guest journey in order to improve the quality and/or efficiency of the intervention, and to ensure the focus of care is geared towards outcomes most valued by guests [34]. Healthcare process or journey mapping is a method used to examine components of healthcare service encounters (for example, activities, interventions, or staff interactions) in order to identify problems or barriers to the provision of efficient, quality care [34, 35]. Focus groups will follow a semi-structured protocol. Interviews will be offered to anyone who cannot or does not want to participate in the focus group.

*Hospital and health services staff.* Online Survey: An online survey will be conducted to explore the feasibility and acceptability of safe space models more broadly, and whether they have an impact on workplace culture and attitudes towards suicide in the broader health system.

Focus Groups or Semi-Structured Interviews: Focus groups will be conducted with teams of hospital and health services staff to explore the acceptability of the safe space models in the broader health system, barriers and facilitators to their adoption, and any impacts the safe space sites have had on workplace culture and attitudes towards suicide. Focus groups will follow a semi-structured protocol with prompts designed to encourage discussion among participants. Interviews will be offered to anyone who cannot or does not want to participate in the focus group.

**Data analysis.** Quantitative data from surveys will be analysed using descriptive statistics to evaluate the views of hospital and health services staff on the feasibility of safe space models. Focus groups/interviews, process mapping, and qualitative data from surveys will use a qualitatively-driven mixed methods approach that incorporates grounded theory strategies with visual representation of the safe space environment, including information on guest and staff movement and interaction, as well as its position within broader healthcare and emergency service systems [34]. Data collection will continue until categories and relationships are saturated and new data do not add to the developing theory [32].

## System and economic study

The system study will use population health datasets and cost data from safe space sites to assess the effectiveness of safe space interventions to reduce presentations for intentional self-harm, suicidal ideation, as well as mental health-related ED presentations.

**Participants and recruitment.** We have requested a waiver of the requirement of consent for the collection of secondary data from Emergency Department datasets on persons who

have presented for suicidal ideation, intentional self-harm, and mental health-related issues in the catchment areas of the safe spaces.

The data to be accessed in this research has been already collected and will be de-identified by the data custodians before it is provided to the project team. Small case cohorts will be represented in any reporting as <5 or, where suitable, aggregated over several years, e.g., 8 incidents between years 2016–2020.

**Data collection.** *Health administrative data*. This study involves secondary data analysis of Emergency Department data from public hospitals in NSW (Emergency Department Data Set), the ACT (Emergency Department Information Service), and SA (Non-Admitted Emergency Care data). The population of interest are those who presented for suicidal ideation, intentional self-harm, or mental health-related issues in the catchment areas of the safe spaces between 2012 and 2024.

Socio-demographic variables, as recorded at the index admission, include: age group (five-year age group); sex; marital status; Local Health District of residence; and country of birth. Visit variables include: Local Health District of facility; arrival date; arrival time; actual departure date; actual departure time; triage category; referral source; reason for presentation; mode of arrival; principal ED diagnosis; referred to on departure; status and place to which person is released; and clinical code set to which a principal diagnosis has been mapped.

In addition to ED data, data from safe space sites will be provided on operating costs (including salaries, wages, overheads, and goods and services costs to run the safe spaces), and guest-reported diversion from ED collected through routine data or the guest online survey. This will be used to evaluate the cost-effectiveness of safe spaces in terms of savings from reduced presentations for intentional self-harm, suicidal ideation, and mental health-related presentations [36].

**Data analysis.** To compare whether presentations for suicidal ideation, intentional self-harm, and mental health-related issues were significantly different before, during, and after implementation of the safe spaces, we will use descriptive statistics to compare prevalence of, and trends in, presentations by age, sex, referral source, mode of arrival, triage category, principal ED diagnosis, and status and place to which person is released. Trends in ED presentation rates by age and sex will be compared with administrative data to evaluate the reach of safe space sites.

To assess the economic impact of safe space interventions on ED presentations and the diversion of ED presentations to safe spaces we will conduct a partial economic evaluation of the safe spaces through cost-benefit analysis from a program perspective. It is difficult to attribute all changes in ED patterns entirely to the safe spaces due to external factors that will also likely influence patterns of ED presentations over the period in question. Moreover, emerging evidence indicates that those attending safe spaces may not have previously accessed health services, including EDs [36]. The economic analysis will thus involve using guest-reported diversion from ED as the primary outcome, supported by a retrospective analysis of all admitted and non-admitted ED separations per patient before and after safe space implementation to estimate the number of avoided presentations. The monetary costs associated with ED-based care will be measured using National Hospital Cost Collection Data and based on average costs per admitted and non-admitted ED separations [37].

## Community study

The community study will be conducted to evaluate the awareness, acceptability, and accessibility of safe spaces to the public and whether those who attend or do not attend these services are representative of the possible population of safe space guests.

**Participants and recruitment.** The community of potential safe space guests includes members of the local communities in which the safe spaces in this project are located. They will comprise health consumers, carers and family members, and other community members. Community members aged 16 years and over who live in a region serviced by a safe space will be recruited. To achieve a broad sample of the local community serviced by each site, the survey will be advertised via social media (e.g., Facebook, Instagram, Twitter, and other relevant websites) tailoring the advertisements to target relevant geographical regions. Flyers and posters with information about the study may also be displayed in relevant community centres and services. We will use the established networks and communication channels of study partners Roses in the Ocean and Suicide Prevention Australia to recruit participants. We will also contact relevant community organisations and peak bodies in the areas serviced by each site and invite them to circulate details of the study via social media, emails, newsletters, and websites. A link to the survey will be provided. Participants will be required to read a participant information statement and to provide informed consent before commencing the survey.

**Data collection.** To elicit the views of community members on the awareness, acceptability, and reach of safe space models we will conduct an online survey with community members who live in a region serviced by one of the safe space sites. The survey aims to measure the awareness and acceptability of the safe spaces and the representativeness of those reached compared with the possible population of guests. The survey will assess participants' knowledge of the safe space in their local region, their likelihood to use the service if needed, and the facilitators and barriers to accessing the service. The survey will also assess participants' likelihood to use other supports and suicide prevention services, any previous experiences of accessing the ED for suicidal crisis or distress, levels of distress, suicidal ideation, and suicidal behaviour, as well as demographic characteristics.

**Data analysis.** Quantitative data from surveys will be analysed using descriptive statistics to evaluate community members' knowledge of their local safe space and the level of need in each region based on reported levels of distress, suicidality, and service use. Qualitative data from surveys will be analysed using framework analysis to assess community members' views on the accessibility and acceptability of the safe space models [38].

## Service co-design study

The service co-design study will evaluate the effects of the co-design process on the development and implementation of individual safe space models, and the successes and challenges of the co-design process that was used to design each of the services. Our lived experience partner organisation, Roses in the Ocean, has been integrally involved in the co-design process of the majority of NSW safe spaces and will facilitate consent to access service co-design documentation and to contact co-design participants for participation in the research.

## Participants and recruitment

Safe space steering committee members, key partners, and co-design participants are those who have contributed to the co-design, set-up, implementation, or managing of the safe spaces. Those who are 16 years and over and who have been active as a key partner, steering committee member, or co-design participant and have contributed to the co-design, set-up, implementation, or managing of a safe space are eligible to participate.

Members of the core project team (key partners) involved in the co-design will be sent details of the online survey and interviews/focus groups. Key partners will be invited to participate in the online survey, interview/focus group, or both. A link to the online survey as well as contact details for the person organising the interviews/focus groups will be provided. We will

also request that they forward this email to steering committee members and co-design participants via their organisational networks.

Participants will be required to read a participant information statement and to provide informed consent before commencing the survey. Participants will be asked to nominate their interest in taking part in an interview/focus group. An embedded link at the end of the survey will take participants to a separate webpage that collects their personal information. Participants will be asked to provide contact details; or alternatively, be asked to contact the listed co-investigator. Alternatively, a recruitment email specifically for the interviews/focus groups will be circulated via the relevant mailing lists, as those who do not participate in a survey may still like to participate in an interview/focus group. Participants will be required to read a participant information statement and complete an informed consent form prior to participating in a research interview/focus group. A reminder email will be sent at 2-weeks if there is no response to the first email. The choice of method will be determined by study participants as part of ongoing co-design.

### Data collection

*Document analysis.* Where these are available and with the consent of the steering committees responsible for producing these documents, we will request access to planning documents developed by key stakeholders as part of the co-design process for individual safe space sites. These documents will provide important insights into the different safe space models across the various trial sites and serve as a resource for evaluating the implementation of these models and their fidelity to the co-design.

*Online survey.* An online survey will be conducted with steering committees, partners, and co-design participants. This will collect data on group formation; meeting processes; communication, decision making and power sharing; personal experiences; and outcomes.

*Focus groups/semi-structured interviews.* Focus Groups or semi-structured interviews will be conducted with safe space steering committee members, partners, and co-design participants. They will follow a semi-structured protocol and will gather information about participants' experiences and perspectives of the co-design process and implementation. Participants' involvement in, expectations of, and outcomes from the co-design process will be a key focus.

### Data analysis

Quantitative data from surveys will be analysed using descriptive statistics to evaluate communication, decision-making, and power-sharing within the co-design process. Qualitative data from surveys and focus groups/interviews will be analysed using framework analysis to systematically map participants' views regarding the personal impact of participating in such a process, the effects on safe space model outcomes, and the successes and challenges of the co-design itself [38]. Framework analysis is a systematic and flexible approach to analysing qualitative data and is appropriate for thematic analysis of textual data where it is important to be able to compare and contrast data by themes across cases, while also situating each perspective in context by retaining the connection to other aspects of each individual's account [38]. Qualitative data from documents will be analysed thematically to understand the co-design process, the context within which the safe spaces were designed, their key features, and as a means of tracking change and development in the implementation of the safe space models [39].

### Data management plan

**Data storage.** The types of research data generated in this project include: survey data, interview data, aggregated health administrative data, documentary data, visual data, audio

files, and metadata (participant consent forms). Data will be stored in a password protected folder on a secure server and only accessible to the research team using individually allocated password protected computers. The project will also generate paper-based records in the form of the evaluation wheel and the graffiti wall collected at specific sites. These data are not identifiable and will be stored securely and only available to project-team members. Hard copy data will be securely shredded immediately after completion of the project.

**Data security and data sharing.**   Arrangements for data custodianship and for members of the research team to access, analyse, and use the data and information has been agreed upon and formalised at the commencement of the project under a Multi-Institutional Agreement. A nominated co-investigator will act as data custodian to facilitate access to the data by other members of the project team while maintaining its protected form. Access to the data will be restricted to those listed as authorised to access the data. Any sharing of data will be through a password protected secure server or through CloudStor folders managed by the project team. Site-specific data, including shared administrative data and uploaded primary data such as digital images, will be managed in individual folders accessible only to relevant site staff and the project team; i.e., site partners will not have access to data from other sites. Only the project team will have access to all folders, and raw data will be protected from alteration by write-protection and restricted access backups.

To negate, minimise, or manage the potential risks from unintended access to data, any direct identifiers will be removed. Any other information that may allow an individual to be identified will be removed or altered so that the risk of an individual being re-identified in the data is very low in the data access environment. Only data necessary for the project member to satisfactorily carry out analyses will be accessible.

## Discussion

Genuine engagement with, and meaningful participation of, mental health consumers, carers, and stakeholders are acknowledged as core principles of co-design and the development of high-quality programs and services [40]. Co-design extends beyond 'consultation' to embed lived experience in the planning, delivery, and evaluation of programs and services, and is underpinned by a focus on early engagement, inclusivity, diversity, transparency, shared power, and equity of knowledge [40, 41]. For research projects, this requires significant commitments of time and resources in building relationships; agreeing on aims, principles, study outcomes, and methods among multiple partners; and working collaboratively to interpret and translate findings for researchers, policy makers, and practitioners [42]. It seems fitting then that a research study exploring the co-creation of safe spaces comprises a significant element of lived experience, with non-academic and non-research-based co-investigators included in the core research team.

Within the process of applying for funding for this co-design project, there were several considerations that may be relevant to the development of future co-designed projects. Placing co-design as central to the grant application involved a degree of risk on the part of the lead investigator and funding body. Embedding co-design across the life of the project requires a level of flexibility and reflexivity within the research process as co-design involves iterative cycles of planning, acting, and reflecting [43, 44]. As such, the funding application had to be sufficiently pliant to allow for processes of ongoing engagement and decision making from stakeholders, as well as the capacity to adjust the research methods as needed [44]. Funding timeframes and priorities, coupled with established university and health systems within which the project must be conducted, place additional demands upon the research team [42]. These may be exacerbated when service user experiences and needs are prioritised within the research design [45].

The emergent and adaptive nature of the co-design approach also presented unique challenges in gaining ethics committee approval [46]. As Goodyear-Smith et al. [46] explain, ethics committees require clear, pre-specified information about the nature and delivery of interventions and how their impacts will be measured. When the focus is on co-creating research through multi-stakeholder partnerships and the continuous quality improvement of service design, such rigid pre-specification is contradictory to the aims and principles of co-design. To this end, the ethics committee application aimed to ensure flexibility and adaptability in the research design to accommodate context-sensitive approaches and feedback processes, while providing significant detail for the purposes of ethical approval [46]. Data collection tools were co-created with people with lived experience of suicide and service providers and contain question sets that illustrate the most intrusive areas of questioning and outcomes of interest, with banks of questions from which the final set(s) will be chosen. These will be developed in collaboration with stakeholders to maximise acceptability at each site. The subset of finalised materials and questions may differ across sites. Modification of approved tools will be submitted to the ethics committee for final approval.

## Safety considerations

It is possible that some participants will experience psychological discomfort in this study. For example, some participants might consider the survey questions sensitive or might find talking about their circumstances upsetting. Information will be provided to participants advising who they can contact, should they feel distressed through any involvement in the study. All information statements include a list of possible sources of help, tailored to the participant group. Online surveys will include a link in the footer to the same list of help sources, with links to relevant sites such as Beyond Blue and Lifeline chat. In-person activities such as body mapping, interviews, and focus groups will be conducted by experienced project staff, who will be trained in the use of the distress protocol. The protocol includes ensuring the availability of peer and clinical staff to support participants who become distressed. The protocol also includes respectfully asking participants to consider someone they would like to contact in the event of distress, which puts the choice of how to address distress in the hands of the participant before data collection commences. A considerable proportion of the guest feedback will be collected while guests are onsite at the safe space with staff available to support guests.

Another potential concern is the participation of safe space staff and key partners, steering committee members, and co-design participants in interviews about their experiences, workplace, or the co-design process. As the study is small, and explores potentially sensitive topics such as workplace culture, leadership, and effective processes, there is the chance that participants may not feel comfortable being frank about their experiences due to the risk of identification. The limited number of participants overall and within each setting means this risk cannot be completely mitigated and this will be clearly communicated to all participants upon entry to the study and before interviews and focus groups commence.

All qualitative data for each setting will be integrated during analysis and every effort will be made to remove comments that may identify individuals. All interview participants will be given the opportunity to review transcripts of their interviews and remove any comments with which they are no longer comfortable. However, this will not be possible for focus groups, where completely redacting comments is often not possible in the context of conversations. Participants will also be given the opportunity to review drafts of outputs where data from their interviews is used. Identification of specific sites in outputs will be discussed with site partners on a case-by-case basis, balancing the need to protect potentially identifiable individuals (e.g., through limited number of positions of that type) with the practicality of masking.

However, as stated above, all participants will be warned that they may still be identifiable despite these efforts. Our previous work in this area suggests that participants who feel well-informed about the potential risks are comfortable taking part and providing honest assessment of their experiences, even when it may not paint a favourable picture of their workplace.

## Limitations

The study has some potential limitations that must be acknowledged. The difficulties of sensitively collecting detailed and rigorous data from people in crisis may result in small sample sizes and potential selection biases and may not be representative of all populations accessing safe spaces. The absence of a comparison group in the administrative datasets limits some of the conclusions that may be drawn about the impacts of the safe spaces, and the challenges of attributing direct population-level outcomes to safe spaces are significant. Limited opportunity to collect long-term outcome data from safe space guests is a further limitation that needs to be acknowledged and impacts our ability to evaluate safe spaces beyond their short-term impacts.

## Strengths of the planned study

The study aims to provide robust, practical research evidence for safe spaces in terms of their effect on ED presentations, alleviation of distress, and linkage to support and services, as well as a greater understanding of the barriers and facilitators to ongoing implementation. This will inform local delivery of safe spaces and provide a broader understanding of the key features of safe spaces as acceptable and effective alternatives to hospital-based care for people experiencing emotional distress and/or suicidal crisis, especially those who might usually present to the ED or choose not to access help through an ED due to past negative experiences.

Recently in Australia there has been a focus on the implementation and evaluation of place-based suicide prevention initiatives, such as LifeSpan, which deliver multiple evidence-based suicide prevention strategies into a community at one time [47, 48]. If found to be effective, safe spaces could be added to the suite of strategies available to communities to tackle suicide and contribute to an evolving system of care for people at risk of suicide that is more empowering, less traumatising, and not just focused on clinical care.

Key stakeholders, particularly those with lived experience of suicide either as members of steering committees, co-design participants, or as suicide prevention peer workers, will play a key role in service design. This study will explicitly investigate the way that service co-design occurred for each service and its influence on implementation. This will provide critical new knowledge on the successes and challenges of co-design to inform future practice, including the impact of factors such as setting, organisational culture, and leadership. It will also contribute to a growing body of research on the role and benefits of peer support by providing greater understanding of how peer work delivered by specialised suicide prevention peer workers can affect peoples' experiences of suicidal crisis or distress.

## Acknowledgments

The following are members of the Stride Safe Space Team: Juliet Middleton, Kate Snars, Marta Jakubczak, Melissa Sum, and all service staff and peer workers.

## Author Contributions

**Conceptualization:** Michelle Banfield, Scott J. Fitzpatrick, Heather Lamb, Melanie Giugni, Alison L. Calear, Erin Stewart, Maree Pavloudis, Lucy Ellen, Ginny Sargent, Helen Skeat,

Bronwen Edwards, Benn Miller, Amelia Gulliver, Louise A. Ellis, Vida Bliokas, Purity Goj, Melissa Lee, Kelly Stewart, Glenda Webb, Merkitta Main, Carrie Lumby, Kelly Wells, Carolyn McKay, Philip J. Batterham, Alyssa R. Morse, Fiona Shand.

**Funding acquisition:** Michelle Banfield, Heather Lamb, Alison L. Calear, Ginny Sargent, Helen Skeat, Amelia Gulliver, Louise A. Ellis, Vida Bliokas, Philip J. Batterham, Alyssa R. Morse, Fiona Shand.

**Methodology:** Michelle Banfield, Scott J. Fitzpatrick, Heather Lamb, Melanie Giugni, Alison L. Calear, Erin Stewart, Maree Pavloudis, Lucy Ellen, Ginny Sargent, Helen Skeat, Bronwen Edwards, Benn Miller, Amelia Gulliver, Louise A. Ellis, Vida Bliokas, Purity Goj, Melissa Lee, Kelly Stewart, Glenda Webb, Merkitta Main, Carrie Lumby, Kelly Wells, Carolyn McKay, Philip J. Batterham, Alyssa R. Morse, Fiona Shand.

**Project administration:** Michelle Banfield, Scott J. Fitzpatrick, Melanie Giugni.

**Supervision:** Michelle Banfield.

**Writing – original draft:** Michelle Banfield, Scott J. Fitzpatrick, Heather Lamb, Alison L. Calear, Erin Stewart, Lucy Ellen, Ginny Sargent, Helen Skeat, Philip J. Batterham, Alyssa R. Morse, Fiona Shand.

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
