## [Editor Report · Decision Letter 0]

21 Jul 2022

Co-creating safe spaces: Study protocol for translational research on innovative alternatives to the emergency department for people experiencing emotional distress and/or suicidal crisis

PONE-D-22-17243

Dear Dr. Fitzpatrick,

We’re pleased to inform you that your manuscript has been judged scientifically suitable for publication and will be formally accepted for publication once it meets all outstanding technical requirements.

Kind regards,

Thomas Phillips, PhD

Staff Editor

PLOS ONE

Journal Requirements:

2. lease provide additional details regarding participant consent. In the ethics statement in the Methods and online submission information, please ensure that you have specified what type you obtained (for instance, written or verbal, and if verbal, how it was documented and witnessed). If your study included minors, state whether you obtained consent from parents or guardians. If the need for consent was waived by the ethics committee, please include this information.

"MB, HL, AC, GS, HS, AG, LE, VB, PB, AM and FS are funded by the Australian Government Department of Health funded National Suicide Prevention Research Fund managed by Suicide Prevention Australia. The funders have had no role and will not have a role in study design, data collection and analysis, decision to publish, or preparation of the manuscript."

"MB, HL, AC, GS, HS, AG, LE, VB, PB, AM and FS are funded by the Australian Government Department of Health funded National Suicide Prevention Research Fund managed by Suicide Prevention Australia. The funders have had no role and will not have a role in study design, data collection and analysis, decision to publish, or preparation of the manuscript."

7. One of the noted authors is a group or consortium Stride Safe Space Team. In addition to naming the author group, please list the individual authors and affiliations within this group in the acknowledgments section of your manuscript. Please also indicate clearly a lead author for this group along with a contact email address.

8. Your ethics statement should only appear in the Methods section of your manuscript. If your ethics statement is written in any section besides the Methods, please move it to the Methods section and delete it from any other section. Please ensure that your ethics statement is included in your manuscript, as the ethics statement entered into the online submission form will not be published alongside your manuscript. 
---

## [Editor Report · Acceptance letter]

26 Sep 2022

PONE-D-22-17243 

Co-creating safe spaces: Study protocol for translational research on innovative alternatives to the emergency department for people experiencing emotional distress and/or suicidal crisis 

Dear Dr. Fitzpatrick:

I'm pleased to inform you that your manuscript has been deemed suitable for publication in PLOS ONE. Congratulations! Your manuscript is now with our production department. 

Kind regards, 

on behalf of

Dr Thomas Phillips 

Staff Editor

PLOS ONE